# Combined Liquid Biopsy Methylation Analysis of CADM1 and MAL in Cervical Cancer Patients

**DOI:** 10.3390/cancers14163954

**Published:** 2022-08-16

**Authors:** Markus Leffers, Johanna Herbst, Jolanthe Kropidlowski, Katharina Prieske, Anna Lena Bohnen, Sven Peine, Anna Jaeger, Leticia Oliveira-Ferrer, Yvonne Goy, Barbara Schmalfeldt, Klaus Pantel, Linn Wölber, Katharina Effenberger, Harriet Wikman

**Affiliations:** 1Institute of Tumor Biology, University Medical Center Hamburg-Eppendorf, 20246 Hamburg, Germany; 2Department of Gynecology, University Medical Center Hamburg-Eppendorf, 20246 Hamburg, Germany; 3Department of Transfusion Medicine, University Medical Center Hamburg-Eppendorf, 20246 Hamburg, Germany; 4Department of Radiotherapy and Radio-Oncology, University Medical Center Hamburg-Eppendorf, 20246 Hamburg, Germany

**Keywords:** liquid biopsy, methylation, CADM1, MAL, cervical cancer

## Abstract

**Simple Summary:**

In cervical neoplasia, tissue methylation of CADM1 and MAL has been identified as a promising triage tool for HPV-positive women. Here, we performed an exploratory study on the potential of the methylated genes as combined markers for the analysis of cervical smears and peripheral blood samples in cervical cancer patients. The combination of both markers in liquid biopsy resulted in a sensitivity of 83.3% and specificity of 95.5% for discrimination of cervical cancer from healthy controls. The robustness of the assay was proven by a respectably high sensitivity in stage I-II patients.

**Abstract:**

Cervical cancer is the fourth most common cancer in women, which is associated in >95% with a high-risk human papillomavirus (HPV) infection. Methylation of specific genes has been closely associated with the progress of cervical high-grade dysplastic lesions to invasive carcinomas. Therefore, DNA methylation has been proposed as a triage for women infected with high-risk HPV. Methylation analyses of cervical cancer tissue have shown that cell adhesion molecule 1 (CADM1) and myelin and lymphocyte protein (MAL) methylation are present in over 90% of all cervical high-grade neoplasias and invasive cervical cancers. Here, we established a liquid biopsy-based assay to detect MAL and CADM1 methylation in cell free (cf)DNA of cervical cancer. Methylation of the target gene was validated on bisulfite converted smear-DNA from cervical dysplasia patients and afterward applied to cfDNA using quantitative real-time PCR. In 52 smears, a combined analysis of CADM1 and/or MAL (CADM1/MAL) showed methylation in 86.5% of the cases. In cfDNA samples of 24 cervical cancer patients, CADM1/MAL methylation was detected in 83.3% of the cases. CADM1/MAL methylation was detected already in 81.8% of stage I-II patients showing the high sensitivity of this liquid biopsy assay. In combination with a specificity of 95.5% towards healthy donors (HD) and an area under the curve (AUC) of 0.872 in the receiver operating characteristic (ROC) analysis, CADM1/MAL cfDNA methylation detection might represent a novel and promising liquid biopsy marker in cervical cancer.

## 1. Introduction

Cervical cancer is the most common cause of cancer-related death in many less developed countries, and worldwide the fourth most common malignancy among women. In 2020, over 340,000 women died of cervical cancer [1]. High-risk human papillomavirus (hrHPV) infection is by far the greatest risk factor for the development of cervical cancer, with >95% of patients being HPV-positive [2]. Although the introduction of HPV vaccines has had a huge impact on hrHPV infection rates, a large part of the human population still has not received the vaccination [3,4,5]. Therefore, longitudinal screening is essential for early detection and surveillance for more effective treatment planning. For this, different liquid biopsy approaches, such as cell free (cf)DNA analyses, have shown great promise as an easily obtainable minimally invasive tool for early detection and disease monitoring [6,7]. In virus-associated cancer, such as Epstein-Barr-virus-related nasopharyngeal carcinomas, the detection of viral DNA fragments in blood has been shown to be a highly specific and sensitive liquid biopsy tool [8,9]. In addition to the detection of circulating HPV-DNA, epigenetic changes can also serve as highly sensitive liquid biopsy biomarkers not hampered by the confounding effects of clonal hematopoiesis [10]. In cervical cancer, specific promoter methylations have been identified as important enabling mediators of how high-grade dysplastic lesions (HSIL/CIN2-3) can progress to invasive carcinomas [11]. Especially, the promoter methylation of the genes cell adhesion molecule 1 (CADM1) and myelin and lymphocyte protein (MAL) has been extensively studied in cervical tissue and smears and could frequently be observed in the development of cervical cancer [12,13]. CADM1 is known as a tumor suppressor gene and is involved in cell-cell interactions and adhesion [14,15]. MAL is also defined as a tumor suppressor gene in many cancers. MAL is implicated in apical sorting and raft stabilization [16]. Diagnostically, combined detection of methylation of both genes in cfDNA could prove to be an easily obtainable potential alternative molecular monitoring tool for hrHPV-positive women in the future [17,18,19,20]. To our knowledge, this is the first liquid biopsy-based study on a combined CADM1 and MAL analysis in cancer patients. The aim of this work was to establish a combined CADM1 and MAL methylation assay suitable for smears and liquid biopsy and to investigate whether this methylation pattern remains carcinoma-specific in cfDNA compared to healthy donors (HD). Furthermore, the first evidence on sensitivity and significance in relation to clinical characteristics as tumor stage has been collected.

## 2. Materials and Methods

### 2.1. Patients and Clinical Characteristics

To validate the methylation assay, cervical smears from 52 patients with an indication for cervical conization were collected and tested (Appendix A). In 42 patients this was due to CIN 3, in 10 due to CIN 2. The average age of the patients was 39.6 years. Thirty-seven (71.2%) patients had previous cytologic findings of Pap IIID2 or more suspicious. Twenty-one (40.4%) patients were positive for one HPV type, twenty-eight (53.8%) for multiple HPV types, and three (5.8%) were HPV negative. HPV type 16 was present in 46.2% of cases, HPV 18 in 9.6%, and HPV 31 in 7.7%. For the evaluation of a liquid biopsy assay, EDTA blood was collected from 30 HD and from 24 cervical cancer patients between May 2019 and March 2021 (Appendix A). Follow-up samples were collected from three cervical cancer patients. The HD group (*n* = 30) included all female blood donors acquired through the Transfusion Medicine of the University Medical Center Hamburg-Eppendorf. The average age of the HD was 38.2 years, and the average age of cervical cancer patients was 46.2 years at the time of blood draw. For standardization, all tumor stages were adjusted to the FIGO classification renewed in 2019 [21]. Four patients were in stage FIGO IA, five in IB, two in IIB, nine in IIIC, and four in IVB. Blood samples were collected in relation to therapy, either before or during therapy. Twelve patients were already receiving tumor therapy at the time of blood collection; of these, ten were receiving chemotherapy or radio-chemotherapy; two had already completed tumor surgery in the therapy regimen and were about to start radio-chemotherapy. Twelve blood samples were drawn from patients before starting therapy. Of these, nine patients were diagnosed for the first time (two of them received their diagnosis during a performed conization) and three patients were about to start a new chemotherapy for progressive disease. Histologically, 22 carcinomas were squamous cell carcinomas and two were adenocarcinomas. In 17 patients, HPV infection was recorded in patient files, one patient was HPV negative, and in six no information was available. All participants gave their written informed consent which was approved by the local ethical committee (No. PV-5392, 06/12/2016, Ärztekammer Hamburg).

### 2.2. Plasma Isolation and DNA Extraction

Plasma was extracted from EDTA blood using a two-stage centrifugation protocol (10 min at 300× *g*, followed by 10 min at 1800× *g*) and was stored at −80 °C until further processing. Two 7.5 ml tubes were collected from cancer patients whereas 1 tube was obtained from HD. All plasma from the blood samples was used for DNA extraction. Extraction was performed using the QIAamp Circulating Nucleic Acid Kit (Qiagen, Hilden, Germany) as described in the manufacturer’s protocol. DNA from the smears collected in SurePath smear emulsion was isolated directly on the day of sample collection. DNA extraction was performed using the QIAmp MinElute Media Kit (Qiagen, Hilden, Germany). The protocol was scaled up to double the volumes for higher DNA yield, allowing 500 µL of SurePath smear emulsion to be used for DNA isolation. DNA concentrations were determined using Qubit 4 (Thermo Fisher Scientific, Waltham, MA, USA) and the samples were stored at −20 °C until further use.

### 2.3. Bisulfite Conversion and Quantitative Methylation Specific PCR (qMSP)

Bisulfite conversion of the extracted DNA was performed using the EZ DNA Methylation-Lightning Kit (Zymo Research, Irvine, CA, USA) according to the manufacturer’s protocol. Based on the highest sensitivity for cervical cancer tissue in literature, the methylation site M12 for CADM1 and the methylation site M2 for MAL were analyzed using the qMSP primers and TaqMan probe sequences as described by Overmeer et al. [22]. As internal reference control, beta-actin (ACTB) was used with primers and the TaqMan [22]. To enable d4uplex analysis, the probes for CADM1-M12 and MAL-M2 were linked with FAM as fluorophore at the 5’-end and BHQ1 as the quencher at the 3’-end; the ACTB probe with 5’-Cy5 and 3’-BBQ (Eurofins Genomics, Ebersberg, Germany). Fully methylated and bisulfite-converted DNA at all CpG sites served as a positive control; DNA was isolated from MDA-MB-231 cells and modified with the methyltransferase M.SssI (New England Biolabs, Ipswich, MA, USA) according to the manufacturer’s protocol [23]. For CADM1-M12, DNA isolated from CAMA-1 cells served as a negative control [24], and for MAL-M2 DNA isolated from primary human keratinocytes [22,25], both bisulfites converted. Non-bisulfite converted MDA-MB-231 DNA treated with M.SssI and a no-template control served as the additional negative controls in each run. The Epitect MethyLight PCR Kit (Qiagen, Hilden, Germany) was used for PCR. The reaction mixture with a total volume of 20 µL contained 10 µL EpiTect MethyLight Master Mix, 2 µL Primer/Probe Mix (CADM1-M12 or MAL-M2 and ACTB; 0.4 µM per primer and 0.2 µM per probe), and 10 ng template DNA. Amplification was performed in a 45-fold, 2-step cycle with an initial activation step at 95 °C for 5 min; followed by 5 s 95 °C denaturation and 60 s annealing and extension at 60 °C for CADM1-M12 analyses and at 59 °C for MAL-M2 analyses. The BioRad CFX96 C1000 Touch (BioRad, Hercules, CA, USA) was used. At 10 ng total DNA input, the dilution series showed a sensitivity of 0.1% for CADM1-M12-qMSP and a sensitivity of 0.2% for MAL-M2-qMSP for a ratio of methylated DNA to non-methylated DNA with an efficiency of 97.5% and 92.1% for CADM1 and MAL, respectively (Figure 1). Each patient sample was tested at least in duplicates, or if possible, in triplicates. The HD samples were tested once (eight samples only for CADM1, twenty only for MAL, and two for CADM1 and MAL) due to the low amount of cfDNA. Only results with an ACTB Ct value less than 33 were included in the analyses. The 2^−ΔΔCt^ method was used for relative quantification. Samples with a 2^−ΔΔCt^ higher than the 90% confidence interval of the control group were defined as positive as it defined the best specificity and sensitivity.

### 2.4. Statistical Analyses

The final statistical analyses were performed using IBM SPSS Statistics 27. The χ^2^-test with a two-sided significance level of *p* < 0.05 was used to determine significant differences between different patient groups in relation to the frequency of methylation. Correlations in ordinal variables (FIGO stage and tumor grade for the cervical cancer cohort; CIN status and Pap-status for the cervical dysplasia cohort) and metric variables (age) to the frequency of methylation were tested using the Spearman-Rho test. Receiver operating characteristic (ROC) analysis was performed for CADM1 and/or MAL (CADM1/MAL) as a combined marker with 2^−ΔΔCt^ value as a test variable to differentiate cervical cancer patients and healthy donors in liquid biopsy samples.

## 3. Results

### 3.1. CADM1 and MAL Methylation Status of Smear DNA from Cervical Dysplasia Patients

To validate the methylation assay, 52 smear samples from cervical dysplasia patients were analyzed first. CADM1 methylation was detected in 38 (73.1%) and MAL methylation in 25 (48.1%) across all cervical dysplasia patients. A combined analysis of CADM1/MAL showed methylation in 45 (86.5%) cases. Classification for the underlying diagnosis and indication for conization (Table 1) resulted in the division into two groups. Ten patients with biopsy-confirmed CIN 2 showed in 60% CADM1, in 40% MAL, and in 90% CADM1/MAL methylation. Forty-two patients with biopsy-confirmed CIN 3 showed in 76.2% CADM1, in 50% MAL, and in 85.7% CADM1/MAL methylation. In this cohort, no significant correlation to clinic pathological factors was found except for Pap-status in MAL (correlation coefficient = 0.309; *p* = 0.026).

### 3.2. Methylation Status of cfDNA Isolated from Blood from Cervical Cancer Patients and Healthy Donors

CADM1 methylation was detected in 18/24 (75%) of the cervical cancer plasma samples, MAL methylation in 10/24 (41.7%), and CADM1/MAL methylation in 20/24 (83.3%) of the samples (Table 2). To verify the absence of CADM1 methylation in HD, ten HD cfDNA samples were tested to underline the previous results published by Rong et al. [26]. None of these samples were positive. Twenty-two HD samples were tested for MAL methylation, and one (4.5%) was positive. The HPV status of this donor was not recorded. In the case of CADM1, the significance level was *p* < 0.001; in case of MAL, *p* = 0.003 (χ^2^-test). ROC analysis for CADM1/MAL as a combined marker resulted in a significant area under the curve (AUC) of 0.872 (95% confidence interval: 0.743–1.000) for differentiation of cervical cancer patients and healthy donors.

### 3.3. cfDNA Methylation Status of Cervical Cancer Patients in Relation to Clinical Characteristics

CADM1 and MAL methylation was detected in seven (63.6%) and three (27.3%) of the 11 non-metastatic cervical cancer patients (FIGO IA-IIB), respectively; nine (81.8%) patients were positive for either MAL and/or CADM1. In this group, 4/11 patients were FIGO stage IA1, and even in these samples methylation of CADM1 and MAL could be detected in the cfDNA (CADM1: 3/4, MAL: 2/4, CADM1/MAL: 4/4). In nine lymph node metastasized patients (FIGO IIIC), CADM1 methylation was detected in 8/9 samples (88.9%), MAL methylation in 5/9 (55.6%), and methylation in CADM1/MAL in 8/9 (88.9%). In four distant metastatic patients (FIGO IVB), 3/4 were positive for CADM1, 2/4 for MAL, and 3/4 for CADM1/MAL (Table 3). Regarding tumor histology, 86.4% of the 19 squamous cell carcinomas were positive for CADM1 and /or MAL (Appendix A). Comparing the methylation rate between patients who were under ongoing therapy and those who were before therapy, 91.7% (11/12) of the “before therapy” group and 75% (9/12) of the “under ongoing therapy” group tested positive for CADM1/MAL (Table 4). In addition, methylation of CADM1/MAL was detected in 50% (1/2), 78.6% (11/14), and 100% (6/6) of the patients with grade 1, 2, and 3 tumors, respectively (Appendix A). The majority of our patients were HPV-positive (17/24), and 94.1% showed CADM1 and/or MAL methylation in their blood (Appendix A). In this small study cohort, no significant correlation to clinic pathological factors was found except for age and MAL (correlation coefficient = 0.476; *p* = 0.019).

### 3.4. Follow-Up of Cervical Cancer Patients

Exemplarily, follow-up samples were collected from three cervical cancer patients and tested for CADM1 and MAL methylation (Figure 2). The first patient (Pat_1) (Figure 2a) suffered from a fulminant course of the disease within nine months after diagnosis with progression of osseous metastases ending with the patient’s death. Before starting a new chemotherapy with paclitaxel one month after diagnosis of progression, the first blood sample was obtained. Here, no methylation of CADM1 or MAL could be detected in the cfDNA. Four months after diagnosis, a progression of the disease was detected by CT scan, therapy was switched to pembrolizumab, and a second blood sample was obtained one month after the start of therapy. CADM1 methylation was detectable, MAL remained negative.

The second patient (Pat_2) (Figure 2b) was diagnosed with local relapse, after which this patient was enrolled in the Keynote-826 study [27]. Prior to initiation of therapy with carboplatin, paclitaxel, bevacizumab, and pembrolizumab or placebo, the first blood sample was obtained. Here, both CADM1 and MAL methylation was detectable. MRI scans performed during the course every two months initially showed stable disease, followed by seven months under partial response. Under this, at month five, therapy was changed by protocol to bevacizumab plus pembrolizumab or placebo. Upon the occurrence of a complication, therapy was changed to solely pembrolizumab. Eleven months after diagnosis of a relapse, a second blood sample was obtained. Here, the methylation status remained unchanged, i.e., positive for CADM1 and MAL methylation. In the subsequent MRI scan at month 12 after diagnosis, progressive disease was stated, and the tumor was regrowing, indicating a constant tumor lead.

The third patient (Pat_3) was also included in the Keynote-826 study on the new-onset distant metastases in lymph nodes of the neck (Figure 2c) [27]. One and a half months after diagnosis (under ongoing chemotherapy with carboplatin, paclitaxel, bevacizumab, and pembrolizumab or placebo), the first blood sample was obtained and both MAL and CADM1 methylation were detectable. During the first 11 months after diagnosis, a tumor status of partial response determined by CT scan was steadily present. Five months after diagnosis (one month after the end of chemotherapy and switch to maintenance therapy with bevacizumab and pembrolizumab or placebo), CADM1 methylation was detectable whereas MAL methylation was no longer detectable. Ten months after diagnosis, both CADM1 and MAL methylation were again detectable in the third blood sample.

## 4. Discussion

Methylation analyses of cervical cancer tissue have shown that CADM1/MAL methylations are present in up to 99% of all squamous cell carcinomas and 92% of all adenocarcinomas [12,22,28]. Furthermore, methylation of both these genes in tumor tissue has been published for several other tumor entities making these highly interesting liquid biopsy markers in general [16,29,30,31,32,33]. Therefore, the main aim of this work was to establish and investigate the combined use of CADM1 and MAL promoter methylation as liquid biopsy markers in cervical cancer with the following goals: (a) to provide the possibility to make a qualitative statement as to whether the sample is methylated or not methylated; (b) to achieve the highest possible sensitivity, and (c) to be specific towards HD samples. To our knowledge, this is the first study showing results on a combined liquid biopsy CADM1 and MAL assay in cancer patients. For assaying cfDNA in plasma, the available amount of circulating tumor (ct)DNA is a major critical limiting factor. Numerous studies have shown that both the total amount of cfDNA as well as ctDNA varies greatly among patients, depending on factors such as age or disease stage [7]. Therefore, we first tested the sensitivity of our liquid biopsy assay using a dilution series starting with 10 ng DNA input, respecting the fact that the five-fold input amount was published before by Overmeer et al. for this assay developed for smear and tissue samples [22]. Overmeer et al. achieved a sensitivity of 0.1% methylated DNA against a background of unmethylated DNA in their dilution series for CADM1 and MAL methylation using qMSP [22]. We achieved a sensitivity for cfDNA-based qMSP of 0.1% and 0.2% for CADM1 and MAL, respectively.

Going into patient material, we chose smears from CIN 2 and 3 patients, to investigate whether the downscaled assay remained similarly sensitive in a patient group comparable to the previously published study [22]. In smears of our CIN 3 patient group, 76.2% were positive for CADM1 and 50.0% were positive for MAL, which is comparable to 66.7% and 53.3% in the previous study [22]. The χ^2^-test did not identify a significant difference between the methylation rates between these two cohorts. From these results, it could be concluded that the assay scaled down to 10 ng DNA input is reliable and sensitive and thus suitable for liquid biopsy approaches.

Our pilot methylation analysis in carcinoma patient plasma shows that a high positivity rate of 83.3% is achieved across all tumor stages by combining both markers. No significant differences in the frequency of methylation with respect to tumor stages were observed. Obviously, more samples are needed to test whether any clinical correlation may exist. Even in very small microinvasive tumors (FIGO IA1) and in patients who were already under therapy, a positive methylation status could be detected in cfDNA. Specificity to cfDNA samples from HD was also shown. Only one HD out of 30 showed positive methylation for MAL (specificity of CADM1 100%; of MAL 95.5%). In a previous study of MAL methylation in liquid biopsies, a 100% specificity was also not achieved [34].

In general, our calculated sensitivity for MAL methylation in plasma alone for the detection of cervical cancer is 41.7%. Guerrero-Preston et al. reported a sensitivity of 94.1% and a specificity of 73.3% in a small set of samples including 17 breast cancer patients and 15 HD [34]; Agostini et al. reported a 13% sensitivity and 100% specificity among 39 breast cancer patients and 49 healthy donors [35]. These results show that a wide range of sensitivity and not necessarily 100% specificity to healthy controls can be obtained, depending on the method used, and the clinical, pathological, and geographical characteristics of the samples analyzed.

For CADM1, one study in cfDNA in cervical cancer has been published [26]. With a cut-off of 1.81 × 10^4^ methylated copies/mL, Rong et al. were able to achieve a sensitivity of 53.6% and a specificity of 91.4% regarding the prediction of metastasis in 14 metastatic and 176 non-metastatic cervical cancer plasma samples [26]. Our calculated sensitivity for CADM1 methylation in plasma alone for the detection of cervical cancer is 75%; the specificity to healthy controls is 100%. In contrast to our study, the previous study found significant differences between lymph node positive and negative as well as metastatic and non-metastatic stages. We could show that CADM1 is already commonly methylated in early-stage tumors. Although a slightly lower level of methylation was found for stage I-II (63%) compared to ≥stage III patients (85%), no significance was obtained (*p* = 0.855). Obviously larger studies are needed to verify these results.

We were able to monitor the treatment of three progressive cervical cancer patients. The results show that although the assay is very robust on its own, the use of the combination of the two methylation markers succeeds over one alone. This was primarily seen for both markers in Pat_2 and Pat_3 while in Pat_1, only one of the markers (CADM1) came up. Missing detection of methylated ctDNA at this “non-tumor-free” disease stage may most likely be caused by a ctDNA amount below our detection limit potentially due to ongoing therapy effects. These promising findings need to be validated in a larger follow-up cohort.

By combining both markers, our assay achieved a sensitivity of 83.3% in detecting cervical cancer while maintaining a high specificity of 95.5% towards HD. Other methylation analyses already performed in liquid biopsy on cervical cancer show similar results compared to ours. Human maternally expressed gene 3 (MEG3) reached a sensitivity of 90.5%; E-cadherin (CDH1) of 42% [36,37]. Regarding the results from methylation analyses, it is worthwhile to compare them also with those of other types of liquid biopsy analyses in cervical cancer. To date, these have mainly involved the analysis of tumor-specific mutations. An analysis of two phosphatidylinositol-4,5-bisphosphate 3-kinase catalytic subunit alpha (PIK3CA) mutations by digital droplet PCR yielded a sensitivity of 22.2% [38]; further analyses by next generation sequencing (NGS) yielded sensitivities ranging between 31–35% for different mutations [39]. In another study, not correcting for possible clonal hematopoiesis mutation rates, up to 83% was reported [40]. The comparison shows that in terms of sensitivity, methylation analyses may be superior to ctDNA mutation analyses. As a limitation of our study, the small and heterogeneous group of cervical cancer patients should be mentioned. Nevertheless, our study shows that the assay seems feasible for multiple stages of this disease. Moreover, further analyses including patients with precancerous lesions should also be performed as methylation of these markers has been shown to also be relevant in pre-cancer stages.

## 5. Conclusions

We have successfully been able to carry out a combined marker methylation analysis in liquid biopsy of cervical cancer patients. With detection limits down to 0.1% (CADM1) and 0.2% (MAL), these markers have the potential to serve as sensitive additional and easily applicable tools in cervical cancer management. Further in-depth and larger studies will be necessary to fully explore the capability of these markers in liquid biopsy in different clinical settings. Monitoring disease progress and therapy response by a simple blood draw seem within reach by subsequent CADM1/MAL methylation marker analysis. This may support gynecologic care of cervical cancer in the future.

## Figures and Tables

**Figure 1 cancers-14-03954-f001:**
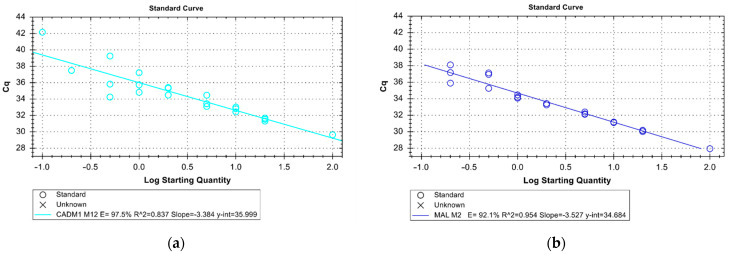
Dilution series of the CADM1 (**a**) and the MAL (**b**) qMSP for assessing the sensitivity and efficiency of the assay. The *x*-axis indicates the starting concentration of fully methylated DNA diluted in unmethylated DNA in percent on a logarithmic scale by a total DNA input of 10 ng. Triplicate analyses were performed for each dilution except for 100%. The *y*-axis shows the Cq at which the threshold value was exceeded (Ct-value). For CADM1, an efficiency of PCR of 97.5%, detection of methylated DNA down to 0.1% is possible. For MAL an efficiency of PCR of 92.1%, detection of methylated DNA down to 0.2% is ensured.

**Figure 2 cancers-14-03954-f002:**
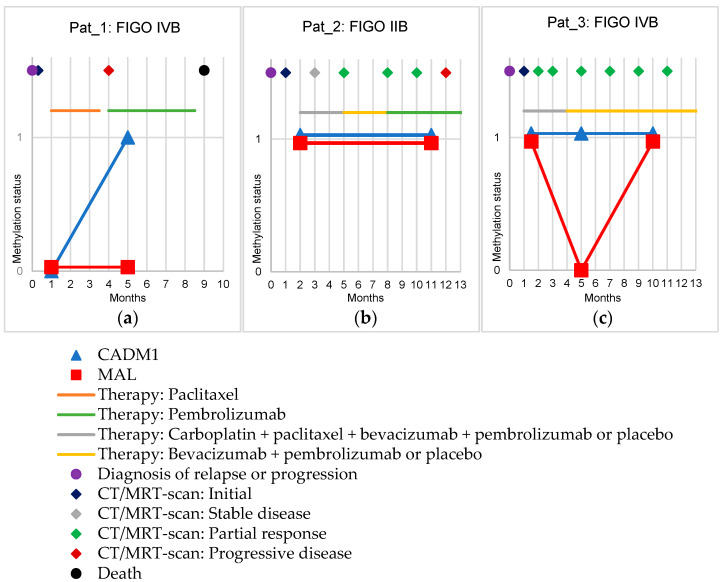
Methylation status of CADM1 and MAL (shown by the *y*-axis: 0 means negative, 1 means positive) in follow-up samples of three cervical cancer patients. The *x*-axis shows the time in months. In addition, therapies (lines) and disease progression (dots and diamonds) are shown. (**a**): Shows the follow-up of a patient in FIGO stage IVB with the progression of osseous metastases. (**b**): Shows the follow-up of a patient in FIGO stage IIB with a local relapse. (**c**): Shows the follow-up of a patient in FIGO stage IVB with a relapse in the form of newly appeared distant lymph node metastases.

**Table 1 cancers-14-03954-t001:** CADM1, MAL, and CADM1 and/or MAL methylation status of smear samples from 52 cervical dysplasia patients in relation to the severity of dysplasia.

	Diagnosis for Surgery Indication			
	CIN 2		CIN 3	
	Count (N = 10)	%	Count (N = 42)	%
CADM1	6	60.0%	32	76.2%
MAL	4	40.0%	21	50.0%
CADM1 and/or MAL	9	90.0%	36	85.7%

**Table 2 cancers-14-03954-t002:** CADM1 and MAL methylation status in blood samples from healthy donors and cervical cancer patients.

		Healthy Donor		Cervical Cancer	
		Count	%	Count	%
CADM1	Negative	10	100.0%	6	25.0%
	Positive	0	0.0%	18	75.0%
MAL	Negative	21	95.5%	14	58.3%
	Positive	1	4.5%	10	41.7%

**Table 3 cancers-14-03954-t003:** CADM1, MAL, and CADM1 and/or MAL methylation status of blood samples from 24 cervical cancer patients in relation to tumor stage (FIGO classification 2019).

	FIGO Stage					
	IA-IIB		IIIC		IVB	
	Count (N = 11)	%	Count (N = 9)	%	Count (N = 4)	%
CADM1	7	63.6%	8	88.9%	3	75.0%
MAL	3	27.3%	5	55.6%	2	50.0%
CADM1 and/or MAL	9	81.8%	8	88.9%	3	75.0%

**Table 4 cancers-14-03954-t004:** CADM1, MAL, and CADM1 and/or MAL methylation status of blood samples from 24 cervical cancer patients in relation to point of blood sampling.

	Point of Blood Sampling			
	Before Therapy ^a^		Under Ongoing Therapy ^b^	
	Count (N = 12)	%	Count (N = 12)	%
CADM1	9	75.0%	9	75.0%
MAL	6	50.0%	4	33.3%
CADM1 and/or MAL	11	91.7%	9	75.0%

^a^ Seven initial diagnoses, two initial diagnoses after conization (local R1), and three relapse diagnoses. ^b^ Five under radiochemotherapy, five under chemotherapy, and two between tumor surgery (local R0) and radiochemotherapy.

## Data Availability

All methods and data have been presented in the paper.

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
