# Peer review of "Combined Liquid Biopsy Methylation Analysis of CADM1 and MAL in Cervical Cancer Patients"

_cancers, 2022, doi:10.3390/cancers14163954_

Round 1

Reviewer 1 Report

I would like to congratulate the authors to performing the study focused on the detection of CADM1/MAL methylation in liquid biopsy samples and testing the usefulness of the method for the monitoring of cervical cancer therapy. The study is well designed and executed. The writing is largely clear and concise in good English style. I appreciate the method by which is possible to extract 10 nanograms of cfDNA.

I have some comments that do not reduce the quality of the study:

- Why authors have chosen the method by Overmeer et al. for detection of CADM1/MAL methylation? There are many other methods useful for detection of methylation of the mentioned genes.

- How authors come to the fact, that 90% confidence interval of control group will be the optimal value for evaluating the positivity of the sample?

- How could authors explain that methylation status of CADM1 and MAL was lower in more serious stages of cervical cancer?

- The results should be validated on a larger number of patients. Are the authors planning to test the method on a larger cohort?

Author Response

Review 1

Comments and Suggestions for Authors

I would like to congratulate the authors to performing the study focused on the detection of CADM1/MAL methylation in liquid biopsy samples and testing the usefulness of the method for the monitoring of cervical cancer therapy. The study is well designed and executed. The writing is largely clear and concise in good English style. I appreciate the method by which is possible to extract 10 nanograms of cfDNA.

  • Many thanks for this positive comment.

I have some comments that do not reduce the quality of the study:

  1. Why authors have chosen the method by Overmeer et al. for detection of CADM1/MAL methylation? There are many other methods useful for detection of methylation of the mentioned genes.
  • Indeed, there are numerous papers using different methods and primers for these two target genes. Our choice of method was because the paper from paper Overmeer et al. was one of the first reporting a significant effect of the combination of markers in cervical tumor tissue. Silencing of genes by methylation is taking place usually at CpG islands. Often different CpG sites have different impact on gene silencing. Thus, we decided to choose those primers targeting specific CpGs showing the highest impact in cervical tissue compared to HD in a large study cohort. Due to the often low level of cfDNA in samples replicates or many different assays are not always possible.

  1. How authors come to the fact, that 90% confidence interval of control group will be the optimal value for evaluating the positivity of the sample? Line 150
  • This value was chosen as it gave the best specificity and sensitivity. This has now been included in the manuscript. Line 150
  •  
  1. How could authors explain that methylation status of CADM1 and MAL was lower in more serious stages of cervical cancer?

 -          This most likely due to pure chance caused by the small sample numbers of analyzed late stage patients.  We have now modified a sentence in the conclusions “Further in-depth and larger studies will be necessary to fully explore the capability of these markers in liquid biopsy in different clinical settings” Lines 324-326.

  1. The results should be validated on a larger number of patients. Are the authors planning to test the method on a larger cohort?
  • Indeed the study is based on a small number of clinical samples. As stated in discussion the main aim of this pilot study was to establish and investigate the combined use of CADM1 and MAL promoter methylation as liquid biopsy markers. Clearly, this study needs to be validated on a larger number of patients. We do also write in conclusions “Obviously, more samples are needed to test whether any clinical correlation may exist” (Line 269). We are continuously collecting more samples for our biobank but in northern Europe the incidence of cervical cancer is luckily rather low and multi-centered studies would thus be best suited.

Reviewer 2 Report

Leffers, M et al., have developed a novel CADM1/MAL methylation analysis to differentiate cervical cancer from normal samples by using liquid biopsy. The capability to use an easy collection method to use these markers in the detection of the disease has a scientific and clinical interest. In addition, the possibility to use these markers as prognostic biomarkers in therapy response is fundamental.  I recommend no major compulsory revisions but have suggested some minor revisions:

Line 83-84 and (Table S1): Please change the term HPV subtype for the term HPV type to avoid any confusion to the reader. According to the HPV genome variation, there are Types, Subtypes, and Variants. Those HPV described in this article are types, example HPV16, HPV 18, etc

Lines 244-245: Can the authors be a bit more clear with the goal "to provide the possibility to make a pure binary statement"

Lines 254-256: " Overmeer achieved a sensitivity of 0.1% methylated DNA....." This information is not presented in the article referenced. Could the authors add the right reference?

Line 272:  The right number of samples analyzed was 32.  Could please the authors correct or clarify?

279-280 Lines: "These results show that depending on the method used there is a wide range..."  There are differences in sensitivity and specificity not only for the method used. Clinical characteristics of the patient and other factors from the host and population studied, etc can influence. 

290-291 Lines: Could please the authors add the p value?   

Author Response

Review 2

 Comments and Suggestions for Authors

Leffers, M et al., have developed a novel CADM1/MAL methylation analysis to differentiate cervical cancer from normal samples by using liquid biopsy. The capability to use an easy collection method to use these markers in the detection of the disease has a scientific and clinical interest. In addition, the possibility to use these markers as prognostic biomarkers in therapy response is fundamental.  I recommend no major compulsory revisions but have suggested some minor revisions:

  • many thanks for the positive feedback

Line 83-84 and (Table S1): Please change the term HPV subtype for the term HPV type to avoid any confusion to the reader. According to the HPV genome variation, there are Types, Subtypes, and Variants. Those HPV described in this article are types, example HPV16, HPV 18, etc

  • Many thanks for this important correction. We have no corrected this issue both in the manuscript (lines 83 and 84) as well as in the supplementary table 1.

Lines 244-245: Can the authors be a bit more clear with the goal "to provide the possibility to make a pure binary statement"

  • We have now clarified the sentence by changing it to “to provide the possibility to make a qualitative statement as to whether the sample is methylated or not methylated”

Lines 254-256: " Overmeer achieved a sensitivity of 0.1% methylated DNA....." This information is not presented in the article referenced. Could the authors add the right reference?

  • This information was taken from the Overmeer et al., 2010 paper as sited. This value of 0.1% was given in the materials and methods in the section “DNA modification and q-MSP analyses” .

Line 272:  The right number of samples analyzed was 32.  Could please the authors correct or clarify?

  • The numbers for HD samples are indeed a bit confusing as we had 22 samples for MAL and 10 for CADM1. However, for 2 HD samples we had enough cfDNA to be used for both MAL and CADM1 and thus the total number of HD samples is 30 and not 32. This information is explained on lines 146-147. The total number of samples was now included on line 88.

279-280 Lines: "These results show that depending on the method used there is a wide range..."  There are differences in sensitivity and specificity not only for the method used. Clinical characteristics of the patient and other factors from the host and population studied, etc can influence. 

  • Indeed this is correct. We have changed the sentence to “These results show that a wide range of sensitivity and not necessarily 100% specificity to healthy controls can be obtained, depending on the method used, and the clinical, pathological and geographical characteristics of the samples analyzed”.

290-291 Lines: Could please the authors add the p value?  

  • We have now added the non-significant p-value.